**Data Availability Statement:** No datasets were generated or analysed during the current study. All

# The mechanism and effectiveness of mindfulness-based intervention for reducing the psychological distress of parents of children with autism spectrum disorder: A protocol of randomized control trial of ecological momentary intervention and assessment

Qi Wang[1], Siu-man Ng[2], Xiaochen Zhou [2]*

1 School of Graduate Studies, Lingnan University, Hong Kong SAR, China, 2 Department of Social Work and Social Administration, The University of Hong Kong, Hong Kong SAR, China

* xczhou@hku.hk

## Abstract

### Background

Existing studies have unequivocally demonstrated that parents of children with autism spectrum disorder (ASD) experience various stressful daily life events and suffer from psychological distress. Mindfulness level is found to be an effective buffer between parents' appraisal of stress life events and psychological distress. However, the mechanism behind the function of mindfulness is unclear, and traditional mindfulness-based intervention (MBI) in experimental settings is not tailored to personal real-life needs. This study proposes to conduct a randomized controlled trial (RCT) to examine the effectiveness of MBI delivered by ecological momentary intervention (EMI) and assessed by ecological momentary assessment (EMA) in changing participants' cognitive appraisal of stressful life events and thus reducing the psychological distress of parents of children with ASD.

### Method

The proposed study will recruit 670 parents of children with ASD and randomly assign them to the intervention and control groups (335 in each group). Participants in the intervention group will install the EMI/A app on their smartphones. In the app, participants can receive practice prompts daily, browse and practice different mindfulness exercises in the intervention library at any time, talk to a virtual counselor (chatbot) when feeling stressed, complete EMA on the cognitive appraisal of life events, and receive the log of change of psychological status daily. Participants in the control group will only receive audio-based standardized mindfulness practice instructions three times per week. The effects of MBI will be assessed based on the EMA data, right after the intervention and at the 2-month follow-up. The

relevant data from this study will be made available upon study completion.

**Funding:** The author(s) received no specific funding for this work.

**Competing interests:** The authors have declared that no competing interests exist.

**Abbreviations:** MBI, Mindfulness-based intervention; ASD, Autism Spectrum Disorder; RCT, Randomized control trial; EMI, Ecological momentary intervention; EMA, Ecological momentary assessment; CFI, Comparative fit index; TLI, Tucker-Lewis index; SRMR, Standardized root-mean-square residual; RMSEA, Root mean square error of approximation.

primary outcome will be participants' psychological distress measured by the depression anxiety stress scale. The secondary outcomes will include participants' subjective well-being, measured by the satisfaction with life scale, and level of resilience, measured by the psychological empowerment scale. Multilevel structural equational modelling will be applied to examine the pathways of the relationship between daily life events, cognitive appraisal, and psychological distress.

## Discussion

The potential benefit of the proposed study is to increase the psychological well-being of parents of children with ASD, and the method may extend to participants suffering from other psychological issues in the future.

**Trial registration:** This trial has been registered on clinicaltrials.gov with the ID of NCT05746468.

## Introduction

### The burden of parents of children with ASD

Autism spectrum disorder (ASD) is defined as a group of complex neurodevelopmental disorders, including autism, Asperger syndrome, pervasive developmental disorder not otherwise specified (PDD-NOS), and other relevant symptoms and conditions [1]. According to a most updated systematic review, the prevalence of ASD among children aged 4 and 8 years in the United States was 1.70 and 1.85 percent, respectively, while the prevalence in Europe ranged from 0.38 to 1.5 percent [2]. Another meta-analysis reported a pooled prevalence of ASD of 0.27% (95% CI: 0.19–0.35%) in Mainland China, Hong Kong, and Taiwan [3]. The General Household Survey in Hong Kong conducted in 2019 and 2020 by the Census and Statistic Department reported that the prevalence rate of autistic spectrum disorder (ASD) among the local population below 15 is estimated at around 1.4% [4].

To take care of children with ASD, parents may suffer from different hardships. A most recent meta-analysis reported that around 31% (95% CI: 24–38%) and 33% (95% CI: 20–48%) of parents of children with ASD suffer from depression and anxiety, respectively [5]. In the Chinese context, where special education service organizations are limited [6], parents have to make much more effort to educate, coach, and support their children and prepare the independent living of their children with ASD compared with other parents [7]. Families of children with ASD may experience more parenting stress than families of children with other disabilities [8]. Parental psychological distress may also reduce the effectiveness of early teaching interventions on their children and increase children's behavior problems [9].

### Sources of the psychological distress among parents of children with ASD

Commonly, parents of children with ASD may face objective burdens, such as disturbed family relationships, constraints in social, leisure, and work activities, financial difficulties, and subjective burden, such as loss of hope, dreams, and expectations, and embarrassment in social situations [10]. A recent systematic review of 24 studies conducted with Chinese parents with children of ASD summarized four main categories of potential life events of parental stress, including (i) cultural factors derived from Confucianism, which mainly lead to the public's supernatural etiological beliefs and public stigma toward ASD [11]; (ii) parents' own

psychopathological symptoms and autistic traits before their child with ASD was born [12]; (iii) the severity of children' symptoms of problem behaviors [13]; and (iv) caregiver burden, including physical, psychological, financial, social and time burden [14].

While these four major categories of life events were found to be related to the psychological distress of parents, there was also evidence that parents would cultivate and develop resilience from these factors. A qualitative interview study with Chinese parents of children with ASD suggested that parents would improve their moral character to achieve inner peace and mental health based on Confucianism teaching [15]. Chinese culture's collectiveness also helped parents seek family and social support, and this support significantly alleviated the psychological distress of parents.

## The transaction theory of stress and coping

The transaction theory of stress and coping [16] provided the theoretical framework for comprehending why stressful life events would behave differently in different parents of children with ASD under similar circumstances, as well as why parents would develop resilience from stressful life events.

Based on the transactional theory, two kinds of cognitive appraisals are identified: primary and secondary appraisals. Primary appraisal refers to individuals' judgment of whether an encounter is irrelevant, benign-positive, or stressful. If the individual adopts the stressful appraisal, it can be perceived in four forms, including harm/loss (damage already exists), threat (anticipated harms or losses), challenge (events that hold the potential of mastery or gain), and irrelevant [17]. Secondary appraisal denotes individuals' judgment concerning what might and can be done. It includes an evaluation of whether a given coping option will accomplish what it is supposed to, that one can apply a particular strategy or set of strategies effectively, and an evaluation of the consequences of using a specific strategy in the context of other internal and/or external demands and constraints [16, 17].

Researchers have found that positive reappraisal comes from positive beliefs, optimism, parents' cognitive reframing of life events, emotional acceptance and understanding, and adaptability toward having a child with ASD served as coping strategies to deal with psychological distress in Chinese parents of children with ASD [18].

## Mindfulness-based intervention (MBI) and cognitive appraisal of daily life events

According to the transaction theory, the shift of individuals' cognitive appraisal from threat to positive involves the process of metacognition, which is a form of awareness associated with mindfulness [19]. Originated from Buddhism, mindfulness, defined as "the awareness that arises from paying attention, on purpose, in the present moment and non-judgmentally" [20], was adopted in psychological intervention by John Kabat-Zinn and its effectiveness in improving psychological well-being has been approved by different meta-analysis [21, 22].

According to Garland [23] and his Mindful Coping Model [19], mindfulness facilitates participants to transit from stress appraisals into the non-evaluative state of mindfulness, which would interrupt participants' automatic emotional reactions and broaden participants' scope of attention to encompass previously ignored contextual information [24]. Subsequently, when participants' habitual responses get disrupted, they will be able to access a broader set of information from which novel situational appraisals can be generated [24]. In a nutshell, mindfulness in this process enhances cognitive flexibility and leaves participants the space for selecting adaptive reappraisals and encouraging productive re-engagement with stressful life events [24].

On the one hand, mindfulness practice may facilitate positive reappraisal, i.e., after practicing mindfulness, individuals would re-construct stressful events as benign, beneficial, and/or meaningful [25] and further lead to reduced distress and improved mental health outcomes [26]. On the other hand, the self-compassion cultivated from mindfulness practice would alter individuals' secondary appraisal and function as a coping strategy for stress reduction [27]. After recurrent engagement of the mindful reappraisal process, the participants' neuroplasticity and certain brain regions may thus be altered for top-down emotion regulation and positive affectivity. Gradually, mindfulness can function as a transformational practice for replacing maladaptive cognitive habits with healthy and adaptive ones [28]. (The framework can be found in S1 Fig).

## MBI and psychological well-being

Besides the theoretical functions of mindfulness, scholars also examined the clinical effectiveness of mindfulness practice in reducing parents' psychological distress. Existing research has provided evidence that mindfulness practice is effective in reducing stress and enhancing psychological well-being in a variety of clinical and non-clinical groups, particularly with individuals under chronic stress conditions, such as chronic disease, burnout, and caring for individuals with chronic conditions [29–31]. Mindfulness-based practice is also found to be associated with neural changes in specific brain regions, with subsequent effects on attention, affective regulation, mood, psychological well-being, and behavior [32, 33].

Adding mindfulness-based components to existing behavioral knowledge resulted in measurable positive changes and reduced parental stress for parents of children with developmental disabilities [34]. There was also specific clinical evidence showing the effectiveness of MBI on parents of children with ASD. The effect size of a mindfulness intervention on the subjective well-being of caregivers of children with ASD was 0.79 in single-group design ($p < 0.01$) and 0.43 in randomized control trials ($p < 0.001$) [35].

## Ecological momentary intervention and assessment (EMI/A)

There are three main research gaps in understanding the mechanisms of psychological distress in parents of children with ASD and the effectiveness of using MBI for reducing psychological distress. First, due to significant recall bias, traditional observational studies cannot accurately quantify the relationship between life events, cognitive appraisal, and psychological distress. Second, the moderating effects of MBI on modifying parents' cognitive appraisal and psychological distress have not yet been explored with a robust methodology. Third, the traditional delivery face-to-face mode of mindfulness, was hard to attract a large population of parents of children with ASD as these parents are already occupied by their duties of parenting children and maintaining the function of the whole family. Therefore, a newly developed ecological momentary intervention (EMI) and assessment (EMA) platform to deliver MBI is worth exploring.

Patrick et al. first raised the term "ecological momentary intervention" in 2005 [36], which was later defined as "treatments that are provided to people during their everyday lives (i.e., in real-time) and in natural settings (i.e., real-world)" [37]. Normally, using EMI means delivering treatment on mobile phones and being implemented standalone or as a supplement to existing treatments. Meanwhile, EMA is defined as a method using wearable electronic devices to repeatedly document and collect human thoughts, emotions, behavior, and physiological states in the natural environment context [38]. As a special observational study technique, EMA can minimize recall bias, maximize ecological validity, help understand the dynamic interplay between events of interest, and enable ecological momentary interventions [39].

There are several benefits of using EMI and EMA. First, the application of EMI enables researchers to reach large populations easily, rapidly, and cheaply [40]. Secondly, using EMI can modify the contents of intervention based on individual and contextual factors that vary across time and to fit individual needs and backgrounds. Thirdly, By using EMI, participants can practice the skills and exercises in real-life settings that fit their day-to-day routine, reducing the burden on parents of children with ASD to change their daily schedule [40]. Lastly, using EMA enables researchers to obtain information with maximized ecological validity and to track behaviors and experiences change over different periods and contexts [41].

Existing research has offered clinical evidence supporting the effects of EMI and EMA in promoting healthy behavior and psychological well-being [40, 42]. Yet, the literature search on PubMed found no prior study examined the effects of MBI using EMI for parents of children with ASD. Only one study protocol was proposed to explore mindfulness intervention and EMI to decrease adolescent stress and anxiety [43]. The knowledge in filling the above research gaps is essential for examining the mechanism between life events, cognitive appraisal, psychological distress, and methods for reducing the psychological distress of parents of children with ASD.

This study aims to develop and implement the MBI using EMI and EMA platform (EMI/A-MBI) and to assess the effectiveness of this newly developed EMI/A-MBI on reducing psychological distress in parents of children with ASD. Specifically, we aim:

1. to identify the daily sources of depression, anxiety, and stress in parents of children with ASD and what are the protective factors for resilience;

2. (comparison of the intervention and control group) to evaluate the effectiveness of EMI-MBI in improving the psychological well-being of parents of children with ASD;

3. (within the intervention group) to explore whether the intensity and frequency of mindfulness practice moderate the participants' cognitive appraisal of daily life events and their psychological distress and mental well-being.

## Research hypotheses

The research hypotheses of this proposed study include the following:

1. The intervention group would show a significant reduction in depression, anxiety, and stress than the control group right after intervention and during follow-up.

2. The intervention group would significantly increase life satisfaction and empowerment more than the control group right after intervention and during follow-up.

3. The intensity and frequency of mindfulness practice would moderate the relationship between participants' cognitive appraisal of daily life events and psychological distress and mental well-being in the intervention group.

## Methods

### Trial design

This research adopts a parallel-armed, randomized controlled trial (RCT) design. The intervention group will receive a combination of (1) a time-based system-triggered EMA, which will collect participants' sources and status of depression, anxiety, and depression in daily life; (2) EMI-delivered MBI, and (3) a follow-up survey of long-term effects of EMI/A-MBI. The participants will complete a baseline questionnaire and then participate in EMI via a

smartphone application (App) for eight consecutive weeks and receive the daily exercise prompts. The EMA will include questions about self-reported depression, anxiety, and depression. After the 8-week EMI/A-MBI, the participants will be invited to complete a post-experimental survey with similar questions in the baseline questionnaire. Two months after completing the EMI/A-MBI, participants will be contacted to complete a telephone follow-up survey with similar questions in the baseline questionnaire. The control group will receive the longitudinal survey exactly the same as the intervention group and 8-week mindfulness-based short messages sent by the research team on a daily basis. The messages will contain audio-based instructions for mindfulness-based practice, which will be the same as the intervention group. The SPIRIT schedule of enrolment, interventions, and assessments can be found in Fig 1 and the flowchart of the design can be found in Fig 2. (The SPIRIT checklist can be found in S1 File).

**Study setting.** This study will be conducted in Hong Kong and recruit parents of children with ASD from local organizations for ASD, such as Autism Hong Kong, Autism Partnership Foundation, or Autismilee. The study sites will be the current living environment of the participants for the participants' convenience.

**Participants.** The inclusion criteria will comprise (1) parents of children with ASD (children aged between 6–18 and diagnosed with different functional levels of ASD by certified psychologists); (2) own a mobile smartphone with internet access; (3) will stay in Hong Kong

| | STUDY PERIOD | | | | |
|---|---|---|---|---|---|
| | Enrolment | Allocation | Post-allocation | | Close-out |
| **TIMEPOINT**\*\* | *-t₁* | **0** | *t₁* | *t₂* | *t₃* |
| **ENROLMENT:** | | | | | |
| **Eligibility screen** | X | | | | |
| **Allocation** | | X | | | |
| **Informed consent** | | X | | | |
| **INTERVENTIONS:** | | | | | |
| *MBI-EMI/A* | | | ●────────● | | |
| *Audio-based MBI* | | | ●────────● | | |
| **ASSESSMENTS:** | | | | | |
| *Demographic data* | | X | | | |
| *Primary outcome variables* | | X | X | X | X |
| *Secondary outcome variables* | | X | X | X | X |
| *Follow-up* | | | | | X |
| *Qualitative interview* | | | | | X |

Notes: -t1, 0, baseline; t1, start of the intervention; t2, 8-week after the start of the intervention; t3: 2-month follow-up

**Fig 1. SPIRIT schedule of enrolment, intervention and assessment.**

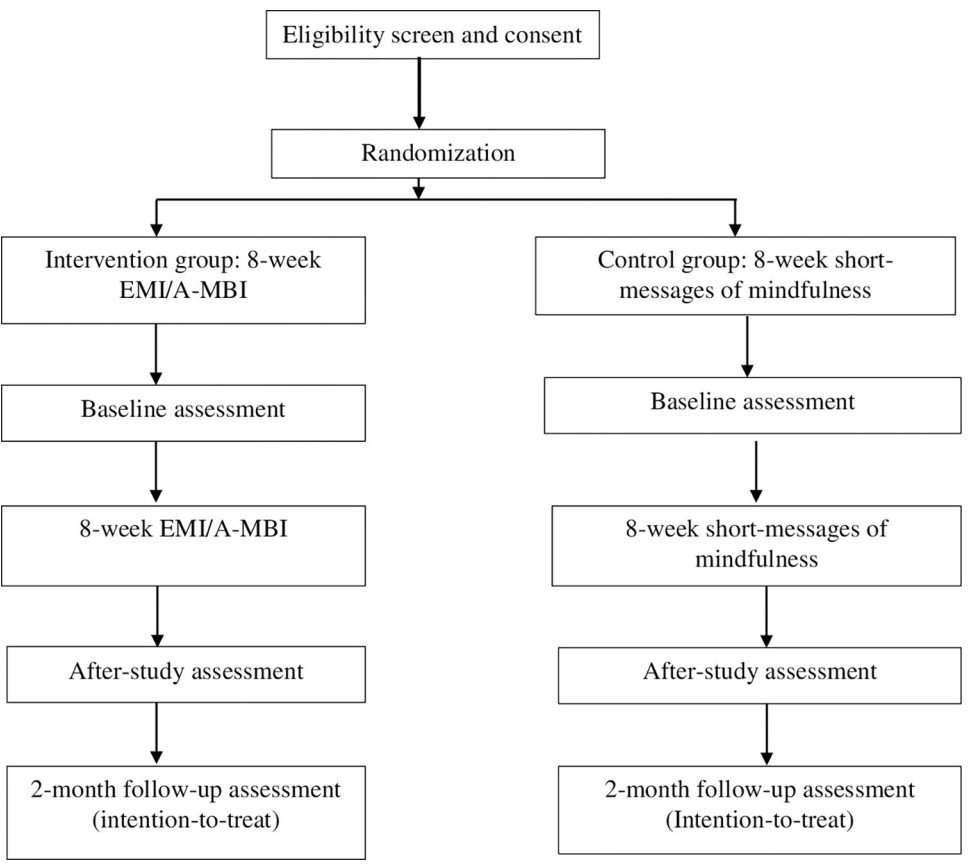

**Fig 2. Flowchart of participants through the trial.**

during the 8-week EMI/A-MBI study period, and (4) able to read and write in Chinese. The exclusion criteria will include (1) parents diagnosed with depression, anxiety, and stress disorder by certified doctors; and (2) parents who do not live together with their children with ASD on a daily basis.

After participants enrol online or are recruited offline from community centers, the research assistant (RA) will explain the study objectives and procedure to participants and get informed consent from all participants.

**Sample size.**   The sample size calculation for this study is based on the linear mixed model that will be adopted to answer research questions 1 and 2 based on the repeated ecological momentary assessment data from the intervention and control group and the effect size of 0.29 from previous studies (unpublished manuscript). The R package *sjstas* [44], and *pwr* [45] were used for sample size calculation. The results indicate that a total of 536 participants will be required to detect a significant difference at an alpha level of 0.05 and a power of 80%. Accounting for potential dropouts, we conservatively estimate that 20% of participants may not complete the study. Therefore, we plan to recruit 670 participants (335 in each group).

**Recruitment.**   Mass emails will be sent to organizations about children with ASD to attract parents interested in the study. Also, online recruitment advertisements will be posted on different online discussion boards. Voluntary participants can directly contact our recruitment staff via telephone, email, or by submitting the online application form for a subsequent briefing meeting. After the briefing session, participants' consent will be obtained.

**Randomization and allocation.**    Individual randomization by a serially numbered opaque sealed envelope (SNOSE) will be used to ensure the allocation sequence was concealed from research assistants and participants before the group allocation. The principal investigator (PI) will prepare about 200 identical, opaque, sealed A5-sized envelopes, with a unique three-digit serial number on the cover of each envelope as an identifier. Half of the envelopes contain an action plan for the intervention group, and the remaining half contains an action plan for the control group. After inserting the intervention materials in the envelopes, they will be sealed, shuffled, and then numbered. When the participant consents to participate, the research assistant opens one envelope according to the serial number sequence and assigns the participant to a treatment condition based on the action plan. All the participants will not be blinded to the given interventions. Outcome assessors (research assistants) are blinded to the group allocation at the telephone follow-up.

**Blinding.**    Although it is not possible to blind the participants to group allocation for psychological interventions, the study purpose will be masked to participants. The study statistician will also be blinded to the group allocation, and participants' personal identifiers will not be included in the dataset.

**Intervention description.**    *EMI/A-MBI app*. The EMI/A-MBI app will include five main parts: a virtual counselor (a chatbot with pre-set algorism for responding to participants), an intervention library, weekly mindfulness practice, an assessment bank, and a daily emotion log. In the eight weeks, the virtual counselor in the chatbot will initiate the conversation every day three times (morning, afternoon, and evening) to check the status of the participants. If the participants responded, the virtual counselor would invite them to rate their depression, stress, and anxiety and then recommend appropriate mindfulness practices in the intervention library, such as 3-minute breathing space or mindful eating. The conversation between the virtual counselor and participants will be in the format of a menu list, and participants can easily choose from different options. After the participants finish the mindfulness practice, the virtual counselor will check the participants' mental health status again. Suppose the participants do not respond to the virtual counselor for the whole day. In that case, the EMI/A-MBI app will prompt a short survey for psychological distress from the assessment bank and recommend mindfulness practice at the end of the day. Participants can also initiate the conversation with the virtual counselor at any time of the day, and the process will be the same. The participants can also go to the intervention library to find the mindfulness practice they prefer. The intervention library will include audio and videos of different mindfulness exercises, such as mindful walking, mindful yoga, or loving and kindness mindfulness. (Example description of the mindfulness practice can be found in S1 Table). Scripts of the audio and videos will also be provided in the intervention library. The daily emotion log will be prompted to the participants every morning since the second day of the intervention as a visual aid for participants to be aware of the change in their psychological distress and mental well-being. Every Saturday, the app will prompt reminders for the participants to join the formal 2.5-hour mindfulness practice. (The App design can be found in S2 File).

**The control condition.**    To control for attention, placebo effects, and the social desirability effect, each participant in the control group will receive three mindfulness practice reminders with audio-recorded instructions on how to practice mindfulness (Audio-Mindfulness). The contents of the audio-recorded mindfulness instructions will be adapted in this study to fit parents of children with autism.

**The explanation for the choice of comparators.**    As mindfulness-based intervention has been proven effective for parents of children with autism, we will involve the components of mindfulness in both the intervention and control groups. Therefore, any extra effects incurred in the intervention group can be attributed to the benefits of using EMI and EMA.

The intervention will be discontinued or modified when participants report feeling more psychological distress while practicing mindfulness-based intervention.

**Relevant concomitant care permitted or prohibited during the trial.** Relevant concomitant care and interventions permitted or prohibited during the trial have been carefully considered following the trial's guidelines. Participants will be allowed to continue any medication or treatment for chronic pain they are currently receiving but will be asked to refrain from beginning any new treatments or therapies during the trial. However, participants will be encouraged to continue their regular daily activities and exercise regimen. To ensure the validity of the intervention, participants will be asked to refrain from engaging in any mindfulness or meditation practices other than those provided by the study during the trial. These measures will contribute to the overall scientific rigor of the trial and ensure that the effects of the mindfulness practice intervention can be accurately assessed.

**Outcomes.** At baseline, post-test, and follow-up survey, the questionnaire contains the same questions concerning demographic information (will only ask at baseline). Psychological distress and mental well-being will be administered to participants in both intervention and control groups. Demographic information, including parent and children's gender, parent and children's age, years of children diagnosed with ASD, number of children living in the same household, family monthly income, parents' education level, parents' employment status, and average time spent with children daily. The primary outcome will be participants' psychological distress measured by the depression anxiety stress scale. The secondary outcomes will include participants' subjective well-being, measured by the satisfaction with life scale, and level of resilience, measured by the psychological empowerment scale. (Detailed measurement information can be found in Fig 3)

*The cognitive Appraisal of Health Scale* will be used to measure participants' cognitive appraisal of daily life events. This scale includes 28 items rated from 1 strongly disagree to 5

| Variables | Measures | Number of questions | Level of measures | Timepoints |
|---|---|---|---|---|
| **Independent variables** | | | | |
| Demographic information | Age (parents and child); gender (parents and child); year when child diagnosed with ASD; family monthly income; employment; education; religious affiliation, hours spend with child with ASD daily; any other family member live together | 11 | | T0 |
| Checklist of risk and protective factors of psychological distress | Modified from the systematic review about factors predicting psychological distress of parents of children with ASD | 8 | Yes (frequency)/No | T1 and anytime participants experienced |
| **Dependent variables** | | | | |
| Participants' cognitive appraisal of daily life events (dependent variable) | The Cognitive Appraisal of Health Scale (modified) | 5 | From 1: strongly disagree to 5: strongly agree | T1 |
| Participants' level of mindfulness | The Five Facet Mindfulness Questionnaire (FFMQ)-short form | 15 at T0, T2 and T3; 5 at T1 | 1: almost always; 2: very frequently; 3: somewhat frequently; 4: somewhat infrequently; 5: very infrequently; 6: almost never | T0 to T3 |
| **Primary outcome** | | | | |
| Participants' level of psychological distress | The Depression Anxiety Stress Scale (DASS) | 21 at T0, T2 and T3; 3 at T1 | 0: Did not apply to me at all; 1: Applied to me to some degree, or some of the time; 2: Applied to me to a considerable degree, or a good part of time; 3: Applied to me very much, or most of the time. | T0 to T3 |
| **Secondary outcomes** | | | | |
| Participants' subjective well-being | Satisfaction with Life Scale | 5 at T0, T2 and T3; 1 at T1 | 7: Strongly agree; 6: Agree; 5: Slightly agree; 4: Neither agree nor disagree; 3: Slightly disagree; 2: Disagree; 1: Strongly disagree | T0 to T3 |
| Participants' level of resilience | Psychological Empowerment Scale for parents of children with a disability | 32 at T0, T2 and T3; 1 at T1 | From 1: never to 5: very often | T0 to T3 |

Note: T0: baseline; T1: ecological-momentary assessment; T2: right after study (after 8-week intervention); T3: follow-up

**Fig 3. Summarizations of measurements.**

strongly agree. This scale was developed based on the transactional theory of stress and coping and thus consists of subscales of primary appraisal (threat, challenge, harm/loss, and benign/irrelevant) and secondary appraisal. The internal consistencies were all greater than 0.70 [46].

*The Five Facet Mindfulness Questionnaire (FFMQ)-short version* [47, 48] will be used to measure participants' level of mindfulness before and after practicing. It includes 15 items measuring the five facets of mindfulness: observing, describing, acting with awareness, non-judging of inner experience, and non-reactivity to inner experience. This scale was further validated [49] and yielded good convergent validity and internal consistency [49].

Parents' psychological distress will be assessed in the domains of depression, anxiety, stress, and mental well-being, indicating participants' life satisfaction and empowerment.

*The Depression Anxiety Stress Scale (DASS)* [50] is a self-report questionnaire that measures the emotional states of depression, anxiety, and stress. This scale contains 21 questions rated from 0 (did not apply to me at all) to 3 (applied to me very much, or most of the time). There were seven questions for each sub-scale of stress, depression, and anxiety [51]. This scale was reliable (alpha = 0.96 and 0.80 for the depression and anxiety subscales, respectively). Cronbach's alpha was high for the three subscales in parents of children with ASD: 0.91 for stress, 0.87 for anxiety, 0.94 for depression, and 0.92 for full scale [51]. It has also been validated in the Hong Kong population [52].

*Satisfaction with Life Scale (SWLS)* [53]. The SWLS has five items scored on a 7-point scale (1 = strongly disagree to 7 = strongly agree) measuring participants' satisfaction with life and has a Cronbach's alpha for internal consistency of 0.87 and a test-retest correlation of 0.82 [53]. This scale has also been validated in Chinese [54] and Hong Kong populations [55].

*Psychological Empowerment Scale for parents of children with a disability (PES)* [56]. It is a 32-item questionnaire on a 5-point scale (1 = strongly disagree to 5 = strongly agree) with four underlying sub-scales: (a) attitudes of control and competence, (b) cognitive appraisals of critical skills and knowledge, (c) formal participation in organizations, and (d) informal participation in social systems and relationships (57). This scale showed high-reliability coefficients (0.90–0.97) [56].

Parents' life events will be prompted daily in the EMI/A app, and questions will be modified from the systematic review that summarizes the risk and protective factors of psychological distress [57]. The shortened version of the scale measuring mindfulness and psychological distress will be embedded in EMI/A and prompt out every day for future analysis.

**Post-trial qualitative interviews.** At the completion of the intervention, participants from the intervention group will be invited for a qualitative interview to evaluate the effectiveness of EMI/A-MBI. Participants who consent to take part in the semi-structured interview will be invited to discuss their experiences of using EMI/A-MBI. All interviews will be audio-recorded. After each interview, digitally recorded audio files will be saved and transcribed by the student helper into text. Nvivo 11 [58] will be used for qualitative data analysis. Significant non-verbal and para-linguistic conversations related to participants' experiences of using EMI/A-MBI will also be noted and recorded. Thematic analysis will be utilized for data analysis, following six steps of thematic analysis, including the familiarity with the data, generating initial codes, searching for themes, reviewing themes, defining and naming themes, and producing the report [59]. The PI and RA will use open- coding to identify relevant content in the transcripts. Then the PI will classify all labeled content into several independent themes showing distinct features of the participants' feedback. Results will then be discussed and consolidated in panel meetings with other co-investigators.

**Follow-up and incentive.** To minimize the survey effect of EMI on the study outcomes and to assess the long-term impact of EMI/A-MBI, a telephone follow-up to assess the study outcomes will be conducted two months after the completion of the EMI/A-MBI.

The EMI validity largely relies on the participants' compliance. Therefore, an attractive incentive is needed. All participants will be given a HK$50 gift voucher upon completing the baseline, post-study, and 2-month follow-up surveys. In addition, participants in the EMI group who have successfully completed all EMIs in the 8-week will be given an additional HK$50 gift voucher.

## Confidentiality

**Ethical consideration and declarations.** This trial was approved by the Human Research Ethics Committee of Lingnan University (Reference No.: EC077/2223, approved protocol can be found in S3 File). Written informed consent will be sought and obtained from all research participants (Example consent form can be found in S4 File), and participants have the right to withdraw from the research at any time without penalty.

The entire protocol will be uploaded on clinicaltrialregistration.org for public access. The participant-level data and statistical code will be available upon request to the corresponding author after the completion of the study.

Personal information about potential and enrolled participants will be collected, shared, and maintained in strict confidence following the SPIRIT guidelines. The SPIRIT guidelines will be accomplished through a variety of measures, including obtaining informed consent from participants, using unique study identification numbers to de-identify data, storing data securely with restricted access, and only disclosing information to authorized parties with a legitimate need to know. In addition, participants will be informed of their rights to access their personal information and withdraw at any time. Before, during, and after the trial, the participants' personal information will be protected by these measures.

## Data management

In consultation with SMN and XCZ, the principal investigator (QW) will be responsible for program coordination and data collection. All digital data, including questionnaires, consent forms, and data from the EMA, will be encrypted and stored on a hard drive following the data management guidelines of Lingnan University. Personal data that can be identified will be stored separately from anonymous data to prevent the identification of personal data throughout the project processing period. The principal investigator will also analyze the data following the guidelines of all phases of data collection, with no interim data analysis. Data access information can be found in data availability statement.

## Data analysis

Data analysis will be conducted using STATA 17 [60]. All significance tests were two-sided with a 5% level of significance. Missing data will be calculated using intention-to-treat approach. Participants who failed to answer the prompted questions will be treat as no improvement on the outcomes. The characteristics of the observed data and the missing data will be compared to determine whether there are any systematic differences. If tested that the missing data will be missing not at random, imputation methods, such as multiple imputation (MI) or maximum likelihood estimation (MLE), will be adopted to estimate the missing values based on the observed data and the relationships between variables. Moreover, sensitivity analysis will be adopted to assess the impact of the missing data on the study results and to determine the robustness of the findings. Complete case analysis and multiple imputation will also be conducted as sensitivity analysis. The internal consistency of the constructs for mindfulness, depression, anxiety, stress, and life satisfaction will be assessed with Cronbach's alpha. In each construct, the PI will sum all items to build the score for that attribute in each participant.

To answer the first and second questions, first of all, the effect size (Heges' g) of the outcome for both the intervention and control groups will be calculated. Subgroup analysis will be conducted when necessary. Next, linear mixed models will be conducted to compare the effectiveness of the mindfulness intervention between the two treatment groups. Linear mixed models are particularly suitable for analyzing data collected via ecological momentary assessment, as they can account for the nested and repeated nature of the data. The main predictor variable will be the treatment group (intervention vs. control), and the model will also include baseline DASS scores and any relevant demographic variables (e.g., age, gender) as covariates. The linear mixed model will include random intercepts and slopes for each participant to account for individual differences in the baseline scores and the rate of change in scores over time. The treatment group effect will be assessed using a fixed effect, and the model will also include an interaction term between treatment group and time to investigate any differential effects of the intervention over time.

To answer the third question, structural equational modelling will be adopted to examine the pathways of the relationship between daily life events, cognitive appraisal, and psychological distress. Moderation analysis will also be used to explore the effects of mindfulness. The model will be adjusted for the participants' role of mother or father. Bootstrapping with size 10,000 will be used to estimate the 95% confidence interval of the parameters. After checking data distribution normality, several fit indices will be served as the judgment of the fit of the model. A non-significant chi-square ($X^2$) is desired, but the test is sensitive to erroneously reject a correct model when the sample size is not large enough [61]. The Comparative Fit Index (CFI) and the Non-Normal Fit Index (also known as the Tucker-Lewis Index, TLI), ranging from 0.00 to 1.00, with higher values representing better fit, will be adopted [62]. The standardized root-mean-square residual (SRMR) and root mean square error of approximation (RMSEA) will be used to detect model misspecifications, with values below 0.08 considered desirable [62]. The indirect effect of the model will be checked with the Sobel test.

## Oversight and monitoring

**Composition of the data monitoring committee, its role, and reporting structure.** A project team will be formed, including all investigators of Lingnan University, HKU, and the agencies where the investigators recruited the participants. The team will meet regularly to design the project's training, intervention, and operation. The authors from the agencies will provide more expert advice on frontline counseling and recruitment, while HKU will provide more consultation on the evaluation and result dissemination.

**Strategies for improving adherence.** After everyday mindfulness practice, participants will be reminded to take the daily mental status assessment via EMA. The measurements will be collected and checked by research assistants via the Dashboard, allowing the RA to track each participant's adherence. For those participants who failed to complete the assessment and practice mindfulness, the RA will send a reminder through WhatsApp.

**Adverse event reporting and harms.** The proposed mindfulness-based intervention is a behavioural approach without invasive procedures and medication; no side effects or adverse outcomes will be expected. All participants will be asked at telephone follow-up by the research assistant if they encounter difficulties using the smartphone app. If so, and the participant agrees, the research assistant will refer these participants to the specialists.

**Plans for communicating important protocol amendments to relevant parties (e.g., trial participants, ethical committees).** Important protocol modifications (e.g., changes to eligibility criteria, outcomes, and analyses) will be reported to the Office of Research Knowledge Transfer, Lingnan University.

**Dissemination plans.**   Upon completion of the proposed research, we shall publish the findings in international peer-reviewed open-access journals and present them in local media or press conferences. The outputs of these result disseminations will be made accessible for public access so that more target beneficiaries can access them. Funding for translating research evidence to practice will be sought to produce educational materials like booklets and videos. These materials will be disseminated to non-profit organizations that serve parents of children with different disabilities. The research team will continue previous academic research and practical collaboration with expertise on mindfulness and keep improving the design of the mindfulness-based intervention.

**Trial status and timeline.**   This trial has been registered on clinicaltrials.gov with the ID of NCT05746468. The tentative timeline of the project is presented in Fig 1. Data recruitment is planned to start on Oct 1, 2023, and will be completed within six months.

## Discussion

This proposed study aims at understanding the mechanisms of psychological distress in parents of children with ASD and using mindfulness-based intervention (MBI) via ecological momentary intervention (EMI) to increase parents' psychological well-being and to further reduce the public health burden. The newly developed eight-week intervention will help parents of children with ASD re-evaluate their current situation, be mindful of daily stressful life events, and further reduce their psychological distress. The reduction of psychological distress would improve parents' physical and mental health.

This trial has both strengths and limitations. First, the strengths include the application of EMI/A-MBI enables the researchers to reach a relatively large population easily, rapidly, and cheaply. The randomized controlled trial study was designed with a large sample size of 670 to ensure sufficient power to examine the efficacy of the EMI/A-MBI. Secondly, using EMI can modify the contents of intervention based on individual and contextual factors that vary across time and to fit individual needs and backgrounds. Thirdly, by using EMI/A-MBI, participants can practice the skills and exercises in real-life settings that fit their daily routine, reducing the burden on parents of children with ASD to change their daily schedule. Lastly, the trial enables researchers to obtain information with maximized ecological validity and track behaviors and experiences change over different periods and contexts.

The trial is not without limitations. First, the validity of the trial largely relies on the participants' compliance to improve which attractive incentives will be provided. Second, the participants in the control group might seek other services or intervention support to reduce their psychological stress, which might challenge the researchers to identify the group difference. We will collect additional data on their participation status in other projects to overcome this. Third, due to the constraints of funding resources, this trial cannot capture the long-term effect of the EMI/A-MBI for parents of children with ASD.

Despite the limitations, the findings of this study may highlight the significance of digital health interventions, such as EMI/A, in the field of mental health care. It may provide evidence for policymakers to support initiatives that educate and train healthcare providers on the use of such tools. This may include funding for research, subsidies for application development, or assistance with integrating these technologies into the existing healthcare infrastructure. In addition, if this study demonstrates the efficacy of MBI, it may be necessary to implement policies to ensure that these interventions are widely accessible. This could involve the incorporation of MBI in health insurance coverage or the development of public programs that serve to disseminate mindfulness techniques to parents of children with ASD and possibly other populations experiencing psychological distress.

## Conclusion

In conclusion, this study aims to shed light on the effectiveness of MBI delivered through digital means, specifically the EMI/A app, in alleviating psychological distress among parents of children with ASD. If the findings are positive, they could signal a turning point in how MBI is delivered, moving away from traditional in-person methods towards more flexible and personalized digital interventions. Ultimately, the potential benefits of this study could extend beyond the participant group to potentially aid individuals struggling with other psychological issues, marking a significant step forward in mental health care.

## Supporting information

**S1 Fig. Conceptual framework of the proposed study.**
(TIF)

**S1 File. SPIRIT 2013 checklist: Recommended items to address in a clinical trial protocol and related documents.**
(PDF)

**S2 File. Preliminary design of the ecological momentary intervention app.**
(PDF)

**S3 File. Research proposal approved by the office of research knowledge transfer, Lingnan University.**
(PDF)

**S4 File. Informed consent form for adult.**
(PDF)

**S1 Table. Example mindfulness practices that will be embedded in the EMI/A app.**
(PDF)

## Author Contributions

**Conceptualization:** Qi Wang, Xiaochen Zhou.

**Funding acquisition:** Qi Wang, Siu-man Ng, Xiaochen Zhou.

**Investigation:** Qi Wang, Xiaochen Zhou.

**Methodology:** Qi Wang, Siu-man Ng.

**Project administration:** Qi Wang, Xiaochen Zhou.

**Resources:** Qi Wang, Xiaochen Zhou.

**Supervision:** Qi Wang, Siu-man Ng, Xiaochen Zhou.

**Writing – original draft:** Qi Wang, Xiaochen Zhou.

**Writing – review & editing:** Qi Wang, Siu-man Ng, Xiaochen Zhou.

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
