## [Decision Letter · Decision Letter 0]

24 Jul 2023

PONE-D-23-10701

The mechanism and effectiveness of mindfulness-based intervention for reducing the psychological distress of parents of children with autism spectrum disorder: A protocol of randomized control trial of ecological momentary intervention and assessment

PLOS ONE

Dear Dr. Zhou,

Thank you for submitting your manuscript to PLOS ONE. After careful consideration, we feel that it has merit but does not fully meet PLOS ONE’s publication criteria as it currently stands. Therefore, we invite you to submit a revised version of the manuscript that addresses the points raised during the review process.

Indicate which changes you require for acceptance versus which changes you recommend

Address any conflicts between the reviews so that it's clear which advice the authors should follow

Provide specific feedback from your evaluation of the manuscript

We look forward to receiving your revised manuscript.

Kind regards,

Syed Far Abid Hossain, PhD

Academic Editor

PLOS ONE

Journal Requirements:

Reviewers' comments:

Reviewer's Responses to Questions

**Comments to the Author**

1. Does the manuscript provide a valid rationale for the proposed study, with clearly identified and justified research questions?

Reviewer #1: Partly

Reviewer #2: Yes

2. Is the protocol technically sound and planned in a manner that will lead to a meaningful outcome and allow testing the stated hypotheses?

Reviewer #1: Yes

Reviewer #2: Yes

3. Is the methodology feasible and described in sufficient detail to allow the work to be replicable?

Reviewer #1: Yes

Reviewer #2: Yes

4. Have the authors described where all data underlying the findings will be made available when the study is complete?

Reviewer #1: No

Reviewer #2: Yes

5. Is the manuscript presented in an intelligible fashion and written in standard English?

Reviewer #1: Yes

Reviewer #2: Yes

6. Review Comments to the Author

You may also provide optional suggestions and comments to authors that they might find helpful in planning their study.

Reviewer #1: In the "Outcomes" section (line 338), it is necessary to provide a clear distinction between primary and secondary outcomes. While the abstract only mentions depression scale as the primary outcome, it is essential to address the inclusion of anxiety and stress as secondary outcomes. The hypotheses and subsequent analyses should align with the designation of primary and secondary outcomes to ensure consistency throughout the study.

The primary aim of the study is to examine whether the intervention reduces psychological distress, as stated in the first hypothesis. Consequently, the primary analyses should focus on comparing the two treatment groups using appropriate statistical tests such as t-tests or linear mixed models. The use of Structural Equation Modeling (SEM) for the first research question is unclear and may not align with the primary outcome of reducing psychological distress.

The sample size calculation should be based on the comparisons between the two treatment groups, rather than derived from SEM considerations. It is crucial to ensure that the sample size is adequately powered to detect differences in the primary outcomes between the treatment groups.

Line 460-462: The analysis procedure described for the second research question is not clearly presented. It is important to note that the effect size is a measure of the difference between two groups and cannot be directly "compared" between the groups.

How will missing data that is not at random (MNAR) be dealt with?

Reviewer #2: The “Introduction” part is lengthy. Most part of it consists of theoretical discussion. There should be a separate “Literature Review” Part. The paper should demonstrate an adequate understanding of the relevant literature in the field and cite an appropriate range of literature sources. The “Methods” part is well-written. The methods are employed appropriately and able to meet the objectives of the paper. The “Results” and “Discussion” parts are well-written. The study should address “policy implications” and “conclusion”.

7. PLOS authors have the option to publish the peer review history of their article (what does this mean?). If published, this will include your full peer review and any attached files.

Reviewer #1: No

Reviewer #2: **Yes: **Sayed Farrukh Ahmed

---

## [Author Response · Author response to Decision Letter 0]

26 Jul 2023

Response to editor

Response: Thank you for your reminder. We have formatted the files following PLOS ONE’s style requirement.

Response: Thank you for bringing this to our attention. Since this is a study protocol and data collection is estimated to take 1-1.5 years, we are unable to provide repository information for the data at this time. However, we plan to release all of our data once the study is complete and manuscript writing has finished.

Response: Thank you so much for your reminder. We have moved the ethics statement at the end of the methods section of the manuscript, above the section on data management. We also deleted the sentence about the ethics statement in the data management section.

Response: Thank you so much for this reminder. We have included a list of the captions of the supporting information files at the end of the manuscript, before the reference section. We also updated all in-text citations to match accordingly. 

Comments to the Author

1. Does the manuscript provide a valid rationale for the proposed study, with clearly identified and justified research questions?

Reviewer #1: Partly

Reviewer #2: Yes

 Response: Thank Reviewer #1’s for the comments. We have made revisions in the method and analysis part to address Reviewer #1’s detailed concerns with the research design. Point-point response is listed below. 

2. Is the protocol technically sound and planned in a manner that will lead to a meaningful outcome and allow testing the stated hypotheses?

Reviewer #1: Yes

Reviewer #2: Yes

3. Is the methodology feasible and described in sufficient detail to allow the work to be replicable?

Reviewer #1: Yes

Reviewer #2: Yes

4. Have the authors described where all data underlying the findings will be made available when the study is complete?

Reviewer #1: No

Reviewer #2: Yes

Response: Thanks for Reviewer #1’s comments. We have restated our data availability at the data management section: Before the conclusion of the research project, no data will be shared, and data will be available upon request to the corresponding author after the completion of the study. Once the study is complete and manuscript writing has finished, we will also upload all of our anonymous data to a data repository website.

5. Is the manuscript presented in an intelligible fashion and written in standard English?

Reviewer #1: Yes

Reviewer #2: Yes

 

6. Review Comments to the Author

Reviewer #1: 

1. In the "Outcomes" section (line 338), it is necessary to provide a clear distinction between primary and secondary outcomes. While the abstract only mentions depression scale as the primary outcome, it is essential to address the inclusion of anxiety and stress as secondary outcomes. The hypotheses and subsequent analyses should align with the designation of primary and secondary outcomes to ensure consistency throughout the study.

Response: In the abstract, we stated that we will utilize the depression anxiety stress scale to assess our primary result. Since this single scale can measure all three factors of depression, anxiety, and stress simultaneously, we have decided to incorporate them as our primary outcome measures. We have added the description of the primary and secondary outcomes in the outcomes section:

“The primary outcome will be participants’ psychological distress measured by the depression anxiety stress scale. The secondary outcomes will include participants’ subjective well-being, measured by the satisfaction with life scale, and level of resilience, measured by the psychological empowerment scale.”

2. The primary aim of the study is to examine whether the intervention reduces psychological distress, as stated in the first hypothesis. Consequently, the primary analyses should focus on comparing the two treatment groups using appropriate statistical tests such as t-tests or linear mixed models. The use of Structural Equation Modeling (SEM) for the first research question is unclear and may not align with the primary outcome of reducing psychological distress.

Response: Thank you so much for pointing out this important issue. We have revised the data analysis section as follows:

“To answer the first and second questions, first of all, the effect size (Heges’ g) of the outcome for both the intervention and control groups will be calculated. Subgroup analysis will be conducted when necessary. Next, linear mixed models will be conducted to compare the effectiveness of the mindfulness intervention between the two treatment groups. Linear mixed models are particularly suitable for analyzing data collected via ecological momentary assessment, as they can account for the nested and repeated nature of the data. The main predictor variable will be treatment group (intervention vs. control), and the model will also include baseline DASS scores and any relevant demographic variables (e.g., age, gender) as covariates. The linear mixed model will include random intercepts and slopes for each participant to account for individual differences in the baseline scores and the rate of change in scores over time. The treatment group effect will be assessed using a fixed effect, and the model will also include an interaction term between treatment group and time to investigate any differential effects of the intervention over time.”

3. The sample size calculation should be based on the comparisons between the two treatment groups, rather than derived from SEM considerations. It is crucial to ensure that the sample size is adequately powered to detect differences in the primary outcomes between the treatment groups.

Response: Thank you so much for your guidance. We have revised the method to calculate sample size and revised it as follows:

 “The sample size calculation for this study is based on the linear mixed model that will be adopted to answer research questions 1 and 2 based on the repeated ecological momentary assessment data from the intervention and control group and the effect size of 0.29 from previous studies (unpublished manuscript). The R package sjstas (45), and pwr (46) were used for sample size calculation. The results indicate that a total of 536 participants will be required to detect a significant difference at an alpha level of 0.05 and a power of 80%. Accounting for potential dropouts, we conservatively estimate that 20% of participants may not complete the study. Therefore, we plan to recruit 670 participants (335 in each group).”

4. Line 460-462: The analysis procedure described for the second research question is not clearly presented. It is important to note that the effect size is a measure of the difference between two groups and cannot be directly "compared" between the groups.

Response: Thank you so much for pointing out this important issue. We have revised the data analysis section as follows:

“To answer the first and second questions, first of all, the effect size (Heges’ g) of the outcome for both the intervention and control groups will be calculated. Subgroup analysis will be conducted when necessary.”

5. How will missing data that is not at random (MNAR) be dealt with?

Response: Thank you so much for pointing out this important issue. We have added in the main text that: 

“The characteristics of the observed data and the missing data will be compared to determine whether there are any systematic differences. If tested that the missing data will be missing not at random, imputation methods, such as multiple imputation (MI) or maximum likelihood estimation (MLE), will be adopted to estimate the missing values based on the observed data and the relationships between variables. Moreover, sensitivity analysis will be adopted to assess the impact of the missing data on the study results and to determine the robustness of the findings.”

 

Reviewer #2: 

1. The “Introduction” part is lengthy. Most part of it consists of theoretical discussion. There should be a separate “Literature Review” Part. The paper should demonstrate an adequate understanding of the relevant literature in the field and cite an appropriate range of literature sources. 

Response: Thank you so much for your suggestion. We have cut some sentences in the introduction part to make it more concise. To follow the manuscript organization requirement of the journal (https://journals.plos.org/plosone/s/submission-guidelines), we kept the current structure of abstract-introduction-method and results-discussion. Thank you for your understanding. 

2. The “Methods” part is well-written. The methods are employed appropriately and able to meet the objectives of the paper. The “Results” and “Discussion” parts are well-written. 

Response: Thank you. We appreciate the compliment.

3. The study should address “policy implications” and “conclusion”.

Response: Thank you so much for your suggestion. We have added the policy implications and conclusion sections at the end of the manuscript on page 23-24.

“Despite the limitations, the findings of this study may highlight the significance of digital health interventions, such as EMI/A, in the field of mental health care. It may provide evidence for policymakers to support initiatives that educate and train healthcare providers on the use of such tools. This may include funding for research, subsidies for application development, or assistance with integrating these technologies into the existing healthcare infrastructure. In addition, if this study demonstrates the efficacy of MBI, it may be necessary to implement policies to ensure that these interventions are widely accessible. This could involve the incorporation of MBI in health insurance coverage or the development of public programs that serve to disseminate mindfulness techniques to parents of children with ASD and possibly other populations experiencing psychological distress.”

“In conclusion, this study aims to shed light on the effectiveness of MBI delivered through digital means, specifically the EMI/A app, in alleviating psychological distress among parents of children with ASD. If the findings are positive, they could signal a turning point in how MBI is delivered, moving away from traditional in-person methods towards more flexible and personalized digital interventions. Ultimately, the potential benefits of this study could extend beyond the participant group to potentially aid individuals struggling with other psychological issues, marking a significant step forward in mental health care.”

---

## [Editor Report · Decision Letter 1]

24 Aug 2023

The mechanism and effectiveness of mindfulness-based intervention for reducing the psychological distress of parents of children with autism spectrum disorder: A protocol of randomized control trial of ecological momentary intervention and assessment

PONE-D-23-10701R1

Dear Author (s),

We’re pleased to inform you that your manuscript has been judged scientifically suitable for publication and will be formally accepted for publication once it meets all outstanding technical requirements.

Kind regards,

Syed Far Abid Hossain, PhD

Academic Editor

PLOS ONE

Additional Editor Comments (optional): Please ensure the format as per Plos One guideline. Copy-edit the manuscript professionally. Thank you. 
---

## [Editor Report · Acceptance letter]

31 Aug 2023

PONE-D-23-10701R1 

The mechanism and effectiveness of mindfulness-based intervention for reducing the psychological distress of parents of children with autism spectrum disorder: 

Dear Dr. Zhou:

I'm pleased to inform you that your manuscript has been deemed suitable for publication in PLOS ONE. Congratulations! Your manuscript is now with our production department. 

Kind regards, 

on behalf of

Dr. Syed Far Abid Hossain 

Academic Editor

PLOS ONE